# Impact of Household Economic and Mothers’ Time Affluence on Obesity in Japanese Preschool Children: A Cross-sectional Study

**DOI:** 10.3390/ijerph20146337

**Published:** 2023-07-10

**Authors:** Kotone Tanaka, Kanami Tsuno, Yasutake Tomata

**Affiliations:** 1Faculty of Health and Social Services, School of Nutrition and Dietetics, Kanagawa University of Human Services, 1-10-1 Heiseicho, Yokosuka 238-8522, Kanagawa, Japan; 2School of Health Innovation, Kanagawa University of Human Services, Research Gate Building TONOMACHI2, 3-25-10 Tonomachi, Kawasaki 210-0821, Kanagawa, Japan

**Keywords:** children, economic status, economic affluence, Japan, obesity, time affluence, time poverty

## Abstract

Although the association between household economic affluence and children’s obesity has been reported, the association between mothers’ time affluence and obesity remains unclear. We conducted a cross-sectional study using Japanese national survey data (2015). The target population was 2–6-year-old preschool children and their mothers. Subjective household economic affluence and mothers’ time affluence were divided into “affluent,” “neither,” “less affluent,” and “non-affluent” groups. Obesity was defined based on the International Obesity Task Force. A logistic regression model was conducted to examine the association between household economic affluence, mothers’ time affluence, and children’s obesity. A total of 2254 respondents were included in the present analysis. The lower household economic affluence was not significantly associated with higher rates of children’s obesity (odds ratio (OR) for the “non-affluent” compared with the “affluent” group was 1.68 (95% CI, 0.93–3.03)). A lower mothers’ time affluence was not significantly associated with higher rates of children’s obesity (OR for the “non-affluent” compared with the “affluent” group was 1.67 (95% CI, 0.92–3.03)). The prevalence of obesity was not synergistically higher when lower household economic affluence and lower mothers’ time affluence were combined.

## 1. Introduction

It has been widely shown that children of low socioeconomic status (SES), primarily those in poverty, are more likely to be obese [1,2,3,4,5,6,7]. Furthermore, childhood obesity is associated with obesity risk in adulthood [8,9], and previous studies have demonstrated a strong, consistent relationship between low SES in early life and increased obesity in adulthood [10,11,12,13,14,15]. Childhood obesity is associated with increased early mortality and risk of later cardiometabolic morbidity (diabetes, hypertension, ischemic heart disease, and stroke) in adult life [16]. Although it could be of great significance to focus on children’s health issues from a life-course perspective, most currently reported studies focus on elementary school age or older, and research on infants and preschool children are still scarce. Indeed, the association between lower household economic status and higher frequency of infant and preschool children’s obesity has been reported in western countries, such as the United States [15,17,18,19,20,21,22], Finland [23], Canada [24], Netherlands [25], while outside of the western countries, there is only one Japanese cross-sectional study [26].

In Japan, income inequality has been increasing since the mid-1980s and is higher than the average defined by the Organization for Economic Cooperation and Development (OECD) [27]. Among the 34 OECD member countries, Japan has the 10th highest child poverty rate, which is also higher than the OECD average. Furthermore, the relative poverty rate of single-parent households is the highest among OECD member countries (data from 2009) [28]. We have previously shown that household economic affluence is associated with obesity among preschool children aged 4 years old [26]. This previous study [26] was conducted using the same questions of household economic affluence from the national representative survey, the National Nutrition Survey on Preschool Children (NNSPC). However, it [26] was conducted in only one region in Japan and only among children attending daycare centers; thus, it may not be a highly representative sample of Japanese preschool children, and research with national representative data is needed.

On the other hand, the concept of “time poverty” has recently been proposed [29,30,31], which suggests that lack of time is an important factor in poverty; lack of time affluence is associated with increased consumption of convenience and pre-prepared foods [30,31] and decreased consumption of home-cooked meals [31]. Adults who work part-time and/or full-time have time-related barriers to healthy eating [32]. Furthermore, working as full-time employees or for long hours can be associated with mothers’ lack of time. It has been suggested that households with mothers who work as full-time employees or for long hours are associated with children having unhealthy eating habits and food intake [33,34,35,36]. These households are also associated with childhood obesity [33,35,36,37,38], and similar results have been reported in Japan [39,40]. Maternal employment is associated with less time spent caring for children and cooking for them; notably, this difference in time (mothers’ working hours and caring for children) is the highest among mothers with young children (0–5 years old) [41]. Therefore, it is expected that the mother’s lack of time affluence due to employment and other factors are associated with children’s obesity through factors such as diet.

In Japan, in recent years, “dual-income households” have been trending upward. The Gender Equality Bureau, Cabinet Office, Government of Japan reported that the number of dual-income households that consisted of an employed husband and a wife without paid work (age of wife ≤64 years) was 718 households in 1985; however, it increased to 1177 households by 2021 (latest data) [42]. Moreover, Japan is a country with a large gender gap, ranking 116th out of 146 countries [43]. According to the OECD, women cared for household members and did routine housework 2.3 times longer than men on average; in Japan, this is 3.7 times longer [44]. Thus, Japanese mothers may tend to have less time affluence. In fact, in our previous study, we found that the effect on child obesity is reinforced when the mother’s lack of time affluence and the household’s lack of economic affluence are combined [26]. However, to the best of our knowledge, a report does not exist that directly examines the impact of mothers’ time affluence on obesity among preschool children.

As described above, there is a lack of research outside of western countries on household economic affluence and children’s obesity, and there is still no research on the impact of mothers’ time affluence on children’s obesity. In addition, the interaction between lack of household economic affluence and lack of mothers’ time affluence is also not yet known. With its relatively high poverty rate among high-income countries and its large gender gap, which makes the lack of time for mothers more problematic, Japan is an ideal country for research on these topics. The present study aims to determine the association between household economic or mothers’ time affluence and obesity in children with national representative data from the NNSPC. We hypothesize that the frequency of obesity would be higher among children with mothers who lacked time affluence and was synergistically higher with the combination of household economic and time affluence (i.e., multiplicative interaction).

## 2. Materials and Methods

### 2.1. Study Design

The present cross-sectional study was conducted using data from the NNSPC 2015 [45]. The NNSPC has been conducted every 10 years since 1985, based on the Statistics Act (General Statistical Surveys, Articles 19–23), to assess methods of feeding in infancy and the diet and lifestyle of preschool children living in Japan, and to obtain primary data required for the planning and promotion of breastfeeding and a healthy diet in early childhood. Potential participants of the NNSPC were children aged <6 years (born from 1 June 2009 to 31 May 2015; approximately 5500 children) and their households (approximately 4400 households) from 1106 districts. They were randomly selected from the districts selected by the National Survey of Living Standards, an institutional statistical survey conducted by the Ministry of Health, Labour, and Welfare of Japan. Trained investigators visited each potential household once in September 2015 and distributed a self-administered questionnaire (for each child) to the mother or guardian who was usually in charge of meal preparation.

### 2.2. Respondents

Of the 2623 respondents, 369 were excluded; hence, 2254 respondents were included in the present analysis. In Japan, men spend 1 h and 54 min per day on housework and childcare, while women spend 7 h and 28 min per day, indicating that the burden of housework and childcare is heavily skewed toward mothers (3.9 times more than men) (*Survey on time use and leisure activities in Japan*, 2021). Therefore, in the present study, we limited the respondents to mothers and examined the association between mothers’ time affluence and children’s obesity. The exclusion criteria were as follows: (1) n = 123: answers not from mothers and (2) missing answers to questions related to (a) n = 6: household economic or mother time affluence; (b) n = 175: children height or weight; (c) n = 5: sex of the child; (d) n = 2: birth weight of the child; (e) n = 15: age of mother; (f) n = 3: care of the child during the day; (g) n = 25: employment status of the mother; (h) n = 11: the number of siblings of the child, and (i) n = 4: the frequency of exercise (Figure 1).

### 2.3. Exposures

Household economic affluence was assessed by asking the question, “How do you feel about your current household economic situation?”, for which available responses were: “most affluent”, “more affluent”, “neither more nor less” (named “neither”), “less affluent”, or “non-affluent.” Mothers’ time affluence was assessed by asking the question, “How do you feel about your current mothers’ time affluence in your life?”, with the same available responses. A total of four groups were created based on the responses, namely, the “affluent” (reference group) that contained response items “most affluent” and “more affluent” and “neither,” “less affluent,” and “non-affluent” as independent groups [26].

To analyze the influence of the combination of household economic affluence and mothers’ time affluence, we created a binary variable where the “affluent” and “neither” was defined as the “affluent or neither” group, and “less affluent” and “non-affluent” were defined as the “not affluent” group. The combination variable of four groups was created: (1) both are “affluent or neither” (=“Yes” and “Yes” in Table 4) (reference group), (2) only mothers’ time is “affluent or neither” (household economic is not affluent) (=“No” and “Yes” in Table 4), (3) only household economic is “affluent or neither” (=“Yes” and “No” in Table 4) (mothers’ time is not affluent), and (4) both are “not affluent” (=“No” and “No” in Table 4). For the sensitivity analysis, we also used a binary variable in which “affluent”, “neither”, and “less affluent” were defined as the “affluent” group, and “non-affluent” was defined as the “not affluent” group. In this analysis, the “affluent” group was set as a reference group.

### 2.4. Outcome

Data on the height and weight of children were obtained from mothers’ reports, and the body mass index (BMI) of children was calculated. Obesity was defined according to the BMI cutoff values for overweight established by the International Obesity Task Force (2-year-old boy: ≥18.13 kg/m^2^, girl: ≥17.76 kg/m^2^; 3-year-old boy: ≥17.69 kg/m^2^, girl: ≥17.40 kg/m^2^; 4-year-old boy: ≥17.47 kg/m^2^, girl: ≥17.19 kg/m^2^; 5-year-old boy: ≥17.45 kg/m^2^, girl: ≥17.20 kg/m^2^; 6-year-old boy: ≥17.71 kg/m^2^, girl: ≥17.53 kg/m^2^) [46].

### 2.5. Confounding Variables

The following items used responses obtained by a mother-reported questionnaire: age (months), sex, birth weight [47,48,49], and mother’s age (years) [50,51,52].

### 2.6. Other Items

The household composition was categorized based on the composition of family members living with children as follows: both of parents (not with grandparents), only father (not with grandparents), only mother (not with grandparents), both of parents (with grandparents), and only mother (with grandparents). The total number of siblings of children was obtained through a mother-reported questionnaire. Lack of exercise was assessed by asking, “How often does your child exercise, including outdoor play and activities at daycare centers, etc.?” The available responses were: “more than 5 days a week”, “3 to 4 days a week”, “1 to 2 days a week”, and “no exercise.” Of these, the largest number of respondents answered, “3 to 4 days a week.” Hence, those who exercised “3 to 4 days a week” or “more than 5 days a week” were defined as the reference, whereas those who exercised “1 to 2 days a week” or “no exercise” were defined as a lack of exercise. Mothers’ occupation was evaluated by analyzing responses to “Are you currently working?” “If yes, what type of work do you currently do?” If the answer was “Not working,” “Full-time employee,” or “Part-time worker”, each was categorized as it is in the response. If the answer was “Contract worker” or “Temporary worker”, they were classified as “Contract or temporary employment”. If the answer was “Directors of companies or organizations,” “Self-employed,” “Family employee,” “Homeworker,” or “Other”, they were classified as “Other”. Information on the main daytime childcare providers was collected by analyzing responses to “Please select all that apply to your child’s primary daytime care provider”. If the answer was “Daycare center,” “Kindergarten,” “Certified childcare center,” “Grandparents or relatives,” “Other,” or “Do not ask for childcare”, each was categorized as it is in the response.

### 2.7. Statistical Analysis

Intergroup comparisons were performed using one-way analysis of variance (ANOVA) for continuous variables and the χ2 test for categorical variables. A logistic regression model was used to calculate multiple adjusted odds ratios (ORs) and 95% confidence intervals (CIs) for obesity according to the household economic affluence categories, with “affluent” used as the reference group. Multivariate models were adjusted for the following variables: Model 1 was not adjusted (crude model); Model 2 was adjusted for children’s age, sex, birth weight, and mother’s age, considering basic individual characteristics. The same analysis was performed for mothers’ time affluence, using “affluent” as the reference group.

In our previous study [26], the risk of childhood obesity was higher only in the “non-affluent” group, suggesting that household economic affluence and childhood obesity may not be a dose–response relationship. Thus, for the combination variable, we defined (1) both are affluent or neither (= “Yes” and “Yes” in Table 4) as the reference group. For the sensitivity analysis of the other combination variable (where only “non-affluent” answers were defined as the “not affluent” group), we also defined (2) both are affluent or neither (= “Yes” and “Yes” in Appendix A Table A1) as the reference group.

Additionally, to test the interaction between the household economic and mothers’ time affluence, we added cross-product terms of these exposure variables (using the binary variables of household economic affluence’s “Yes” or “No” and mothers’ time affluence’s “Yes” or “No”) to Model 2.

We performed the statistical analyses using R version 4.2.1. We considered two-sided *p* values < 0.05 as statistically significant.

## 3. Results

### 3.1. Respondents’ Characteristics

The mean (standard deviation; SD) age of the children was 50.9 (13.4) months (a minimum of 27 months and a maximum of 74 months), and the percentage of boys was 51.6%. The mean (SD) age of mothers was 35.5 (5.2) years. The number of obese children was 138 (6.1%).

Table 1 shows the characteristics of the respondents categorized by the four household economic affluence groups. In the category of household composition, the “non-affluent” group had a higher percentage of single-mother households, a larger number of siblings, a higher percentage of lack of exercise, a lower percentage of full-time employees, and a higher percentage of “do not ask for childcare” than the other (“affluent”, “neither’, and “less affluent”) groups. The characteristics of the respondents categorized by four mothers’ time affluence groups are shown in Table 2. In the “non-affluent” group, a higher percentage of single-mother households and a larger number of siblings were observed, which was similar to that of household economic affluence; however, a lower percentage of lack of exercise were opposite to that of household economic affluence, a higher percentage of full-time employees, a lower percentage of “not asking for childcare”.

### 3.2. Affluence (Household Economic and Mothers’ Time) and Children’s Obesity

The results of the association between household economic or mothers’ time affluence and obesity in children are shown in Table 3. In both the crude model (Model 1) and the model after adding adjusted items (Model 2), household economic affluence was not significantly associated with the prevalence of obesity. The multivariate-adjusted OR in Model 2 for “non-affluent” parents compared to “affluent” parents was 1.68 (95% CI, 0.93–3.03), with no statistical significance, whereas the OR of obesity in children among parents in the “non-affluent” group tended to be higher (*p* for trend = 0.268).

In both Model 1 and 2, mothers’ time affluence was not significantly associated with the prevalence of obesity. The multivariate-adjusted OR in Model 2 for “non-affluent” parents compared to “affluent” parents was 1.67 (95% CI, 0.92–3.03). Additionally, low mother’s time affluence was significantly and inversely associated with the prevalence of obesity in Model 1 (*p* for trend = 0.047) but not in Model 2 (*p* for trend = 0.054).

### 3.3. Combination of Affluence (Household Economic and Mothers’ Time) and Children’s Obesity

The prevalence of children’s obesity, according to the combination of affluence (household economic and mothers’ time), is shown in Table 4. Compared with respondents who either had household economic affluence or mothers’ time affluence, those who had neither household economic affluence nor mothers’ time affluence showed a higher prevalence of children’s obesity, although the results were not significant (multivariate-adjusted OR, 1.38; 95% CI, 0.88–2.15).

As a sensitivity analysis, the prevalence of children’s obesity according to the combination of household affluence (household economic or mothers’ time, classified with “non-affluent” as affluence = “No” and not “non-affluent” = affluence “yes”) is shown in Appendix A Table A1 The prevalence of obesity in children without both household economic affluence and mothers’ time affluence was not significantly higher. Similarly, a test of the interaction between household economic affluence and mothers’ time affluence showed no significant association (*p*-interaction = 0.57).

## 4. Discussion

In the present cross-sectional study, we investigated the association between household economic or mothers’ time affluence and obesity in preschool children. In an analysis of the present study, a lack of household economic affluence was not significantly associated with obesity in children; however, the OR in the “non-affluent” group tended to be higher compared to that in the “affluent” group. These results suggest that a severe lack of household economic affluence tended to be associated with obesity in children. Furthermore, the lack of mothers’ time affluence was not significantly associated with obesity in children, although lower mothers’ time affluence tended to be inversely associated with the prevalence of obesity. This study is the first to suggest that a lack of mothers’ time affluence might result in children’s obesity.

In our previous study [26], the OR of children’s obesity in the household economic “non-affluent” group compared with the “affluent” group was 2.31; however, in the present study, it was 1.68. There are two possible explanations for these discrepancies. First, there is a possibility of a difference in the study population. The study population of the previous study included only children attending a daycare center, whereas the present study included a nationally representative sample (not only including children attending a daycare center but also children attending a kindergarten, being cared for by grandparents or relatives, and others). The utilization rate of daycare centers for 4-year-old in Japan was 42.2% in 2018, according to the latest available data (reported by the Ministry of Education, Culture, Sports, Science, and Technology in Japan, 2021), which is consistent with the results of the present study (40.9%). The Act of Child Welfare Law in Japan ensures that daycare centers in Japan provide care for infants and toddlers who need it for various reasons, such as having working parents. Therefore, the participants in the previous study were more likely to be less affluent in their lives than those in the present study. However, the percentage of obese children in each category of household economic affluence (“affluent”, “neither”, “less affluent,” and “non-affluent”) in the previous study were 6.2%, 7.3%, 5.4%, and 11.5%, respectively, whereas, in the present study, they were 5.6%, 6.2%, 5.7%, and 9.0%, respectively. Since there was not much difference in each group, it is unlikely that there was an underestimation due to selection bias. Second, there is a possibility of larger random errors due to the use of mothers-reported children’s weight and height in this study. Previous studies have reported that mothers are more likely to overestimate their children’s height and underestimate their weight [53]. Thus, underestimation of the association due to a larger random error (non-differential misclassification) might have occurred because the previous study used measurements of weight and height taken by the daycare centers’ staff.

The results of the present study on household economic affluence are logically consistent with previous findings. It is well known that children with low SES, especially those living in poverty, are more likely to be obese [1,2,3,4,5,6,7]. However, the factors that influence household SES on child obesity are not yet clear. The theoretical framework [54] suggests that SES may influence obesity through dietary and exercise behavior that changes energy consumption, energy expenditure, and metabolism. Indeed, the results of the present study also showed that the rate of the lack of exercise among children tends to be higher the lower household economic affluence, and the previous systematic reviews [55] have shown that lower parental socioeconomic status is associated with higher consumption of sugar-sweetened beverages and energy-dense foods in children. It is, therefore, possible that parental SES influences child obesity through diet and other factors, but further research is needed.

We observed that lower mothers’ time affluence tended to be inversely associated with the prevalence of obesity in the present study. Although we have not found any previous studies directly examining the association between mothers’ lack of time affluence and children’s obesity, the results of the present study on mothers’ time affluence may be logically consistent with previous findings. For example, the previous cohort study showed that households with mothers who were employed as full-time workers were associated with a higher BMI and thus had children with excess weight [37]. Furthermore, lack of time affluence is associated with increased consumption of convenience and pre-prepared foods [30,31] and decreased consumption of home-cooked meals [31]. Adults who work part-time and/or full-time have time-related barriers to healthy eating [32], and home-cooked meals using healthy cooking methods reduce the risk of obesity [56] and tend to align with healthier eating patterns [57]. Moreover, female sex, time availability, and employment have also been identified as major determinants of home-cooked meals [58]. The results of the present study showed that the percentage of full-time employees was higher in participants who responded “not affluent” of mothers’ time affluence. The interpretation that mothers’ lack of time affluence is associated with childhood obesity would not be logically inconsistent if full-time employment makes mothers lose time affluence more easily. Furthermore, lower maternal time affluence was found to be associated with a higher proportion of single-mother families and a higher number of children. It is possible that lower mothers’ time affluence is more likely to lead to a decrease in the total amount of care provided to each chid due to the mother’s reduced mental capacity and less time spent on housework and childcare. The mothers’ lack of time affluence may affect their children’s obesity via unhealthy dietary habits.

In the results of the combination of affluence (household economic and mothers’ time) and children’s obesity, the OR tended to be higher when both household economic and mothers’ time affluence were “No”, but it was not statistically significant. The results of the test of interaction effects were also not significant. A previous study reported that mothers working for long hours were a risk factor for overweight and obesity among preschool children, but this was mainly found in high-income families [33,38]. In fact, the subject characteristics of the present study also indicated that the background of the lack of household economic affluence and the lack of mothers’ time affluence may be different. Thus, the risk factors for obesity due to lack of household economic affluence and the risk factors for obesity due to lack of time affluence might be different.

### 4.1. Strengths

This is the first study to directly evaluate the association between mothers’ time affluence and their children’s obesity. The results of the present study were based on relatively representative, governmental, and nationwide survey data.

### 4.2. Limitations

First, the results of the present study might be underestimated due to selection bias, although a nationally representative sample was used. Previously [59], it has been reported that individuals with lower annual household incomes have lower response rates to the NNSPC; this association persisted after adjusting for potential confounders. Thus, the results of the present study might be underestimated because we focused on parents with “less-” or “non-affluence”. The prevalence of overweight has also been reported to be underestimated [59], which may also be a cause of underestimation. Second, since all variables in the present study were subjective data, it is possible that a certain amount of random error (non-differential misclassification) may affect the results. In addition, the present study used only subjective data on time affluence and was not able to examine actual time data. Future research using objective data is required. Third, the present study only included mothers, while fathers and grandparents were excluded from the analysis because the NNSPC did not include data on fathers’ or grandparents’ time affluence. In the previous study, both mothers’ and fathers’ long working hours were associated with overweight young children [38]. Therefore, in the future, it will be necessary to collect information from both mothers and fathers.

## 5. Conclusions

The frequency of obesity tended to be higher among children in households lacking economic affluence and with mothers who lacked time affluence. However, this frequency was not synergistically higher when household economic affluence and mothers’ time affluence were combined. Future research using objective data on household economic affluence and time affluence is required. In addition, mediators, including nutritional factors, need to be clarified in future studies. It might be possible to develop measures to improve the issue of childhood obesity by analyzing and managing mediators.

## Figures and Tables

**Figure 1 ijerph-20-06337-f001:**
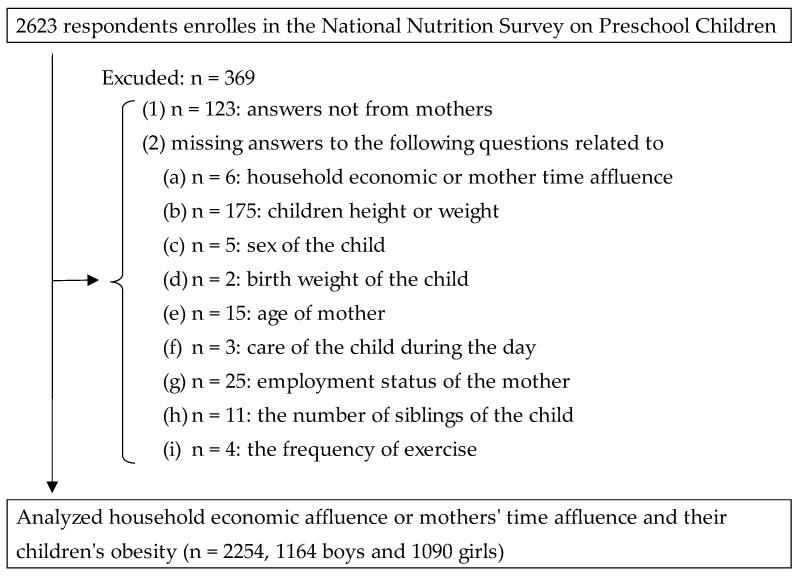
Respondents’ flowchart.

**Table 1 ijerph-20-06337-t001:** Characteristics of the four household economic affluence groups.

	Household Economic Affluence				
	Affluent	Neither	Less Affluent	Non Affluent	*p*-Value
No. of children, n	643	759	652	200	
Mother’s time affluence, n (%)					<0.001
Affluent	312 (48.5)	213 (28.1)	144 (22.1)	29 (14.5)	
Neither	87 (13.5)	246 (32.4)	139 (21.3)	24 (12.0)	
Lessaffluent	198 (30.8)	247 (32.5)	298 (45.7)	88 (44.0)	
Non-affluent	46 (7.2)	53 (7.0)	71 (10.9)	59 (29.5)	
Children’s age (months), mean (SD)	50.3 (13.0)	51.3 (13.6)	51.1 (13.7)	51.1 (13.6)	0.5
Children’s sex (boy), n (%)	326 (50.7)	403 (53.1)	340 (52.1)	95 (47.5)	0.51
Children’s BMI, mean (SD)	15.4 (1.4)	15.5 (1.3)	15.6 (1.4)	15.5 (1.4)	0.42
Children’s Obesity, n (%)	36 (5.6)	47 (6.2)	37 (5.7)	18 (9.0)	0.33
Children’s birth weight, mean (SD)	2989.0 (436.7)	2993.6 (486.2)	2999.2 (433.8)	2967.7 (451.5)	0.86
Mother’s age (years), mean (SD)	36.0 (5.0)	35.7 (5.2)	35.0 (5.1)	35.4 (5.9)	0.006
Household composition, n (%)					<0.001
Both of parents (not with grandparents)	511 (79.5)	546 (71.9)	497 (76.2)	149 (74.5)	
Only father (not with grandparents)	8 (1.2)	6 (0.8)	4 (0.6)	2 (1.0)	
Only mother (not with grandparents)	7 (1.1)	28 (3.7)	25 (3.8)	16 (8.0)	
Both of parents (with grandparents)	97 (15.1)	154 (20.3)	108 (16.6)	22 (11.0)	
Only mother (with grandparents)	20 (3.1)	25 (3.3)	18 (2.8)	11 (5.5)	
Children’s total number of siblings, mean (SD)	1.0 (0.8)	1.2 (0.8)	1.2 (0.9)	1.3 (1.0)	<0.001
Children’s lack of exercise, n (%)	25 (3.9)	38 (5.0)	38 (5.8)	13 (6.5)	0.32
Mothers’ occupational status, n (%)					0.33
Not working	296 (46.0)	329 (43.3)	268 (41.1)	88 (44.0)	<0.001
Full-time employment	167 (26.0)	125 (16.5)	92 (14.1)	17 (8.5)	
Part-time employment	97 (15.1)	209 (27.5)	223 (34.2)	68 (34.0)	
Contract or temporary employment	19 (3.0)	26 (3.4)	20 (3.1)	6 (3.0)	
Other	64 (10.0)	70 (9.2)	49 (7.5)	21 (10.5)	
Main daytime childcare providers, n (%)					
Daycare center	263 (40.9)	303 (39.9)	278 (42.6)	79 (39.5)	0.735
Kindergarden	249 (38.7)	295 (38.9)	233 (35.7)	71 (35.5)	0.531
Certified childcare center	38 (5.9)	55 (7.2)	40 (6.1)	13 (6.5)	0.751
Grandparents or relatives	29 (4.5)	43 (5.7)	32 (4.9)	10 (5.0)	0.8
Other	18 (2.8)	11 (1.4)	5 (0.8)	6 (3.0)	0.02
Do not ask for childcare	71 (11.0)	89 (11.7)	92 (14.1)	32 (16.0)	0.15

**Table 2 ijerph-20-06337-t002:** Characteristics of the four mothers’ time affluence groups.

	Mother’s Time Affluence				
	Affluent	Neither	Less Affluent	Non Affluent	*p*-Value
No. of participants, n	698	496	831	229	
Household economic affluence, n (%)					<0.001
Affluent	312 (44.7)	87 (17.5)	198 (23.8)	46 (20.1)	
Neither	213 (30.5)	246 (49.6)	247 (29.7)	53 (23.1)	
Lessaffluent	144 (20.6)	139 (28.0)	298 (35.9)	71 (31.0)	
Non-affluent	29 (4.2)	24 (4.8)	88 (10.6)	59 (25.8)	
Children’s age (months), mean (SD)	51.3 (12.9)	50.4 (13.8)	50.9 (13.9)	51.3 (12.9)	0.71
Children’s sex (boy), n (%)	359 (51.4)	252 (50.8)	444 (53.4)	109 (47.6)	0.44
Children’s BMI, mean (SD)	15.4 (1.3)	15.6 (1.3)	15.5 (1.5)	15.5 (1.3)	0.15
Children’s Obesity, n (%)	33 (4.7)	31 (6.2)	56 (6.7)	18 (7.9)	0.25
Children’s birth weight, mean (SD)	2969.9 (473.6)	3020.3 (437.9)	2990.3 (452.5)	3000.3 (433.7)	0.30
Mother’s age (years), mean (SD)	35.4 (5.2)	34.9 (5.2)	35.8 (5.1)	36.3 (5.2)	0.001
Household composition, n (%)					0.002
Both of parents (not with grandparents)	548 (78.5)	370 (74.6)	626 (75.3)	159 (69.4)	
Only father (not with grandparents)	7 (1.0)	3 (0.6)	6 (0.7)	4 (1.7)	
Only mother (not with grandparents)	17 (2.4)	14 (2.8)	25 (3.0)	20 (8.7)	
Both of parents (with grandparents)	109 (15.6)	90 (18.1)	144 (17.3)	38 (16.6)	
Only mother (with grandparents)	17 (2.4)	19 (3.8)	30 (3.6)	8 (3.5)	
Children’s total number of siblings, mean (SD)	1.0 (0.8)	1.1 (0.9)	1.2 (0.9)	1.4 (0.9)	<0.001
Children’s lack of exercise, n (%)	34 (4.9)	26 (5.2)	48 (5.8)	6 (2.6)	0.28
Mothers’ occupational status, n (%)					<0.001
Not working	425 (60.9)	208 (41.9)	282 (33.9)	66 (28.8)	<0.001
Full-time employment	49 (7.0)	81 (16.3)	194 (23.3)	77 (33.6)	
Part-time employment	143 (20.5)	150 (30.2)	251 (30.2)	53 (23.1)	
Contract or temporary employment	11 (1.6)	11 (2.2)	37 (4.5)	12 (5.2)	
Other	70 (10.0)	46 (9.3)	67 (8.1)	21 (9.2)	
Main daytime childcare providers, n (%)					
Daycare center	207 (29.7)	201 (40.5)	383 (46.1)	132 (57.6)	<0.001
Kindergarden	343 (49.1)	179 (36.1)	269 (32.4)	57 (24.9)	<0.001
Certified childcare center	40 (5.7)	43 (8.7)	48 (5.8)	15 (6.6)	0.15
Grandparents or relatives	21 (3.0)	23 (4.6)	58 (7.0)	12 (5.2)	0.005
Other	13 (1.9)	7 (1.4)	15 (1.8)	5 (2.2)	0.89
Do not ask for childcare	94 (13.5)	64 (12.9)	106 (12.8)	20 (8.7)	0.30

**Table 3 ijerph-20-06337-t003:** Relationship between household economic or mothers’ time affluence and children’s obesity.

	Household Economic Affluence				*p-Trend*
	Affluent	Neither	Less Affluent	Non Affluent	
No. of children’s obesity, n (%)	36 (5.6)	47 (6.2)	37 (5.7)	18 (9.0)	
Model 1 ^a^	1.00 (ref) ^c^	1.11 (0.71–1.74)	1.01 (0.63–1.63)	1.67 (0.92–3.01)	0.268
Model 2 ^b^	1.00 (ref)	1.11 (0.71–1.74)	1.02 (0.63–1.63)	1.68 (0.93–3.03)	0.262
	**Mothers’ time affluence**				
No. of children’s obesity, n (%)	33 (4.7)	31 (6.3)	56 (6.7)	18 (7.9)	
Model 1 ^a^	1.00 (ref) ^c^	1.34 (0.81–2.22)	1.46 (0.94–2.27)	1.72 (0.95–3.12)	0.047
Model 2 ^b^	1.00 (ref)	1.31 (0.79–2.17)	1.45 (0.93–2.27)	1.67 (0.92–3.03)	0.054

a: Model 1 was a crude model. b: Model 2 was adjusted for children’s age, sex, birth weight, and mother’s age. c: Odds ratios (95% confidence interval).

**Table 4 ijerph-20-06337-t004:** Relationship between the combination of affluence (household economic or mother’s time) and children’s obesity.

Affluent		No. of Children, n	No. of Children’s Obesity, n (%)	Model 1 ^a^	Model 2 ^b^
Economic	Time				
Yes	Yes	858	47 (5.5)	1.00 (ref) ^c^	1.00 (ref)
No	Yes	336	17 (5.1)	0.92 (0.52–0.16)	0.91 (0.51–1.61)
Yes	No	544	36 (6.6)	1.12 (0.78–1.91)	1.22 (0.78–1.90)
No	No	516	38 (7.4)	1.37 (0.88–2.13)	1.38 (0.88–2.15)

a: Model 1 was a crude model. b: Model 2 was adjusted for children’s age, sex, birth weight, and mother’s age. c: Odds ratios (95% confidence interval).

## Data Availability

This data was provided by the Ministry of Health, Labor, and Welfare in Japan.

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
