# Peer review of "Impact of Household Economic and Mothers’ Time Affluence on Obesity in Japanese Preschool Children: A Cross-sectional Study"

_ijerph, 2023, doi:10.3390/ijerph20146337_

Round 1

Reviewer 1 Report (Previous Reviewer 1)

Dear Authors, 

I'm still not convinced with using meta-analysis in this study. The term "meta-analysis" is confusing. Did you mean "comparison" with other studies? Even though this is the aim of the study, it would be helpful to clarify the rationale of using this method. Meta-analysis is a statistical analysis used to quantitively synthesizes studies to derive conclusions about that body of research. However, I haven't seen this used in this study.

Please justify well why men were excluded from the anlaysis in the method/limitation? How many men were excluded? Were they treated as missing values?

Were these covariates used in other studies? Authors just listed these covariates without clarifying their important. For example, why would grandparents/siblings be associated with household economic affluence and childhood obesity?

Moderate editing of English language is required.

Author Response

RESPONSE TO REVIEWER 1:

We sincerely appreciate the time you have taken to carefully review our manuscript and provide valuable input.

  1. I'm still not convinced with using meta-analysis in this study. The term "meta-analysis" is confusing. Did you mean "comparison" with other studies? Even though this is the aim of the study, it would be helpful to clarify the rationale of using this method. Meta-analysis is a statistical analysis used to quantitively synthesizes studies to derive conclusions about that body of research. However, I haven't seen this used in this study.

Response: We thank the reviewer for this comment. The purpose of the meta-analysis in the present study is to obtain more valid point estimates, to reduce the possibility of chance, and to confirm the concordance (heterogeneity) of the association with a previous study. And as we have responded in previous peer reviews, the following meta-analyses, for example, have similar objectives to this study.

1) Borgi L, Rimm E B, Willett W C, Forman J P. Potato intake and incidence of hypertension: results from three prospective US cohort studies BMJ 2016; 353: i2351 doi:10.1136/bmj.i2351

2) Hu Y, Ding M, Sampson L, Willett W C, Manson J E, Wang M et al. Intake of whole grain foods and risk of type 2 diabetes: results from three prospective cohort studies BMJ 2020; 370 :m2206 doi:10.1136/bmj.m2206

3) Zhou, J., Kelsey, K. T., Giovannucci, E., & Michaud, D. S. (2014). Fluid intake and risk of bladder cancer in the Nurses' Health Studies. International journal of cancer, 135(5), 1229–1237. https://doi.org/10.1002/ijc.28764

However, as the reviewer pointed out, the position of the meta-analysis in the present study was not clear, so the meta-analysis was included as a result of the sensitivity analysis and the manuscript was revised to reflect this.

Abstract: Removed description of meta-analysis.

Line225: To evaluate the heterogeneity of the results, a meta-analysis was conducted as a sensitivity analysis to examine the association between household economic af-fluence and children's obesity based on the combined results of our previous study.

Line274: As a sensitivity analysis, The result of the meta-analysis of the association be-tween household economic affluence and children's obesity is shown in Appen-dix Table 1. Meta-analysis of two studies including 4102 individuals showed that household economic affluence was significantly associated with the preva-lence of children’s obesity, and the OR of children's obesity among the not “non-affluent” group to those in the “non-affluent” group was 1.95 (95% CI, 1.273.00). Heterogeneity between the studies was 0%.

Line311-321: Removed description of meta-analysis.

  1. Please justify well why men were excluded from the anlaysis in the method/limitation? How many men were excluded? Were they treated as missing values?

Response: As shown in the flowchart (Figure 1), respondents other than mothers (n = 123) are excluded and not included as missing values. If “men” meant fathers, 33 of the 123 respondents were answered by fathers. The present study only included mothers, and fathers and grandparents were excluded from the analysis because the NNSPC did not include data on fathers' or grandparents’ time affluence. Therefore, we have added the following sentence to the limitation as follows.

Line425: Third, the present study only included mothers, and fathers and grandparents were excluded from the analysis because the NNSPC did not include data on fathers' or grandparents’ time affluence.

  1. Were these covariates used in other studies? Authors just listed these covariates without clarifying their important. For example, why would grandparents/siblings be associated with household economic affluence and childhood obesity?

Response: Covariates, that were not adjustment variables, were listed in out manuscript to consider the background of household economic affluence or mothers' time affluence. However, as the reviewer pointed out, there was a lack of consideration of these background. Therefore, we have added the sentences as follows.

Line243: In the category of household composition, the “non-affluent” group had a high-er percentage of single-mother households, a larger number of siblings, a higher percentage of lack of exercise, a lower percentage of full-time employees, and a higher percentage of “do not ask for childcare” than the other (“affluent”, “nei-ther’, and “less affluent”) groups.

Line249: In the “non-affluent” group, a higher percentage of single-mother households and a larger number of siblings were observed, which was similar to that of household economic affluence; however, a lower percentage of lack of exercise were opposite to that of household economic affluence, a higher percentage of full-time employees, a lower percentage of “not asking for childcare”.

Line330: The utilization rate of daycare centers for 4-year-olds in Japan was 42.2% in 2018, according to the latest available data (reported by Ministry of Education, Culture, Sports, Science and Technology in Japan, 2021), which consistent with the results of the present study (40.9%).

Line365: Indeed, the results of the present study also showed that the rate of the lack of exercise among children tends to be higher the lower household economic af-fluence and the previous systematic reviews [57] have shown that lower paren-tal socio-economic status is associated with higher consumption of sugar-sweetened beverages and energy-dense foods in children.

Line391: Furthermore, lower maternal time affluence was found to be associated with a higher proportion of single-mother families and a higher number of children. It is possible that lower mothers' time affluence is more likely to lead to a decrease in the total amount of care provided to each chid due to the mother's reduced mental capacity and less time spent on housework and childcare.

Line404: In fact, the subject characteristics of the present study also indicated that the background of the lack of household economic affluence and the lack of moth-ers' time affluence may be different.

Reviewer 2 Report (Previous Reviewer 3)

Impact of Household Economic and Mothers’ Time Affluence 2 on Obesity in Japanese Preschool Children: A Cross-sectional 3 Study

REVIEW

Summary

This study was to determine if time affluence is associated with obesity in Japanese preschool children.

Obesity is not significantly associated with mother’s  lower time affluence. Also, when combined, obesity is not higher with lower household economic affluence and mother’s time affluency.

Strength

·       Total of  2,623 respondents of which  369 were excluded. This is a robust database to give the study enough statistical power .

·       A detailed statistical analysis of the data

Weakness

·       The level of  time affluence was defined qualitatively not quantitatively .

·       A better approach will  be  specifying the number of hours spend at home for child care which might give a better quantitative data on time affluence

·       Can it be determined that all the weights and heights provided by the respondents were accurate ?

Recommendations

·       Please can you include a definition of “time affluence” in your discussion.  

·       Minor revisions 

Adequate 

Author Response

RESPONSE TO REVIEWER 2:

We sincerely appreciate the time you have taken to carefully review our manuscript and provide valuable input.

Summary

This study was to determine if time affluence is associated with obesity in Japanese preschool children. Obesity is not significantly associated with mother’s  lower time affluence. Also, when combined, obesity is not higher with lower household economic affluence and mother’s time affluency.

Strength

  • Total of 2,623 respondents of which  369 were excluded. This is a robust database to give the study enough statistical power .
  • A detailed statistical analysis of the data

Weakness

  • The level of time affluence was defined qualitatively not quantitatively .
  • A better approach will be  specifying the number of hours spend at home for child care which might give a better quantitative data on time affluence
  • Can it be determined that all the weights and heights provided by the respondents were accurate ?

Recommendations

  • Please can you include a definition of “time affluence” in your discussion.

Response: We thank the reviewer for this comment. As mentioned the reviewer’s recommendations, we have added the sentences as follows.

Line391: Furthermore, lower maternal time affluence was found to be associated with a higher proportion of single-mother families and a higher number of children. It is possible that lower mothers' time affluence is more likely to lead to a decrease in the total amount of care provided to each chid due to the mother's reduced mental capacity and less time spent on housework and childcare.

Round 2

Reviewer 1 Report (Previous Reviewer 1)

While the authors have made some changes to the manuscript, they have failed to adequately address my underlying concern. Sensitivity analysis in meta-analysis can be used to explore the impact of using different meta-analysis models, or the impact of excluding/including studies based on variance and methodological quality. Authors failed to apply sensitivity analysis strategies. They only used one study to compare with. The distribution of the studies and their confidence intervals around the combined estimate were not presented in a suitable graph. Good quality meta-analysis needs good quality literature searches, and accurate reporting of these searches. Analysis of the studies has to be made to look for possible duplicate publication of the same data. Heterogeneity is measured by using a Cochran’s chi-squared test. This approach is limited when the number of studies is large as heterogeneity measured by a simple chi-squared test is often observed. This is largely because of the high power to be able to detect even small degree of heterogeneity with a large sample size. On this basis, I would recommend declining to publish this work.

Minor editing of English language required.

Author Response

RESPONSE TO REVIEWER 1:

We sincerely appreciate the time you have taken to carefully review our manuscript and provide valuable input.

While the authors have made some changes to the manuscript, they have failed to adequately address my underlying concern. Sensitivity analysis in meta-analysis can be used to explore the impact of using different meta-analysis models, or the impact of excluding/including studies based on variance and methodological quality. Authors failed to apply sensitivity analysis strategies. They only used one study to compare with. The distribution of the studies and their confidence intervals around the combined estimate were not presented in a suitable graph. Good quality meta-analysis needs good quality literature searches, and accurate reporting of these searches. Analysis of the studies has to be made to look for possible duplicate publication of the same data. Heterogeneity is measured by using a Cochran’s chi-squared test. This approach is limited when the number of studies is large as heterogeneity measured by a simple chi-squared test is often observed. This is largely because of the high power to be able to detect even small degree of heterogeneity with a large sample size. On this basis, I would recommend declining to publish this work.

Response: We thank the reviewer for this comment. In response to a reviewer's suggestion, we have removed all descriptions that refer to meta-analysis as follows.

Line98: We also aimed to conduct a meta-analysis to integrate the result of the previous study [28] in order to estimate more plausible point estimates.

Line220: A previous study we conducted [28] on 1848 4-year-old preschool children used the same questionnaire from the NNSPC employed in the present study [28]. In our previous study, household economic affluence was classified using the same definition as in the present study. The definition of obesity was also the same as in the present study, which was defined based on the International Obesity Task Force. To evaluate the heterogeneity of the results, a meta-analysis was conducted as a sensitivity analysis to examine the association between household economic affluence and children's obesity based on the combined re-sults of our previous study. We calculated the OR of children's obesity among the “affluent” group to those in the “non-affluent” group because, in the previ-ous study, the OR was higher in the group with “non-affluent”. Information re-garding effect size and 95% CI from each group was extracted to calculate the OR and 95% CI by random effect model (Restricted Maximum Likelihood (REML)). The heterogeneity of the studies was evaluated using I2 statistics.

Line274: As a sensitivity analysis, The result of the meta-analysis of the association between household economic affluence and children's obesity is shown in Ap-pendix Table 1. Meta-analysis of two studies including 4102 individuals showed that household economic affluence was significantly associated with the preva-lence of children’s obesity, and the OR of children's obesity among the not “non-affluent” group to those in the “non-affluent” group was 1.95 (95% CI, 1.273.00). Heterogeneity between the studies was 0%.

Line349: The OR was higher, especially in the “non-affluent” group in the adjusted model in the previous study (OR, 2.31 [95% CI, 1.234.33]). Conversely, the OR was higher in the “non-affluent” group in the present study (1.68 [0.93-3.03]). Thus, a meta-analysis focused on the prevalence of children’s obesity in the “non-affluent” group compared to the “affluent” was conducted to evaluate the heterogeneity. As a result, we found a significant association between house-hold economic affluence and children’s obesity (OR, 1.95 [95% CI, 1.27-3.00]). The heterogeneity of the results of the meta-analysis was 0%. This suggests that the results of the present study would be considered consistent with those of the previous study [28].

Line457: Appendix Table 1. Meta-analysis of the association between household economic affluence and children's obesity.

We hope that the revised manuscript is suitable for publication in your journal. Thank you in advance for your consideration.

This manuscript is a resubmission of an earlier submission. The following is a list of the peer review reports and author responses from that submission.

Round 1

Reviewer 1 Report

·         Title: Please use “and” instead of “or”.

·         Title: Suggest- in Japanese preschool children.

·         Line 12-13: It is unclear how household economic and mothers’ time affluence were used? Did the authors combine them in one variable? Household economic should be clearly defied in abstract.

·         Line 15: What do you mean by “meta-analysis”? This is a research study not a review paper. Please be aware of using this term throughout the paper.

·         Line 21-23: Please include the OR and P-value only.

·         In abstract, please mention the type of design (e.g., cross-sectional, longitudinal) and the age of participants. Also, the conclusion is highly unacceptable and it does not propose a clear direction for future studies.

·         Please include “Japan” in the keywords list.

·         The introduction is too brief and should be strengthened with more references. There are only 13 references with 19 references used for the whole paper. The section needs to be expanded in my opinion. What is the labour force participation rate of women in Japan? The literature should also be in more depth and discuss the associations between work status and mother's and children’s dietary intake/obesity. For example, working part-time and/or full-time were associated with time-related barriers to healthful eating among adult men/women (Am J Health Behav. 2012 Nov; 36(6): 786-796) and their children (International Journal of Consumer Studies 2018, 42(5): 522-532).

·         Line 40-41: Please explain how mothers’ time affluence is a significant mediating factor between SES and health?

·         Line 41-46: Is there any evidence in Japan or other Asian countries?

·         Line 47-51: The novelty of this paper is very weak in my opinion. The associations between household economic affluence and children's obesity should be clarified by providing more evidence from other countries. Why this study is important to conduct in Japan? Why not in other Asian countries? Why preschool children?

·         Line 69-74: Please describe in much more details how children were recruited?

·          Line 78-82: This is not a good reason for exclusion men from analysis. This needs to be mentioned as a limitation of the study.

·         Figure 1 should be re-organized.

·         Line 106-109: This is not true. Why affluent and neither are combined in one group? This is not logic in my opinion.

·         Line 130: A large number of covariates were used, so there is a high change for bias during analysis.

·         Line 202: than the "affluent" group?

·         I believe Chi-square and ANOVA test should be used in Table 1 & 2. Please include an additional column to show a significant P-value between the four groups.

·         Please combine Table 3 & 4 in one table.

·         Line 265-276: This section doesn’t add any meaningful value to the paper. Please delete.

·         Line 278-287; 324-327: Please do not repeat results in the discussion.

·         The discussion is short and very weak in its current format. I suggest the authors to link the literature with the important results of this study. The implications of results and the directions for future research should be clearly discussed in conclusion.

·         Line 327-331: Delete.

·         Line 343-350: This section should be expanded. Please add additional limitations related to the study methodology and the statistical analysis used.          

·         The manuscript requires significant editing by a native English speaker.       

Author Response

RESPONSE TO REVIEWER 1:

We sincerely appreciate the time you have taken to carefully review our manuscript and provide valuable input.

1) Title: Please use “and” instead of “or”.

Response: As per the suggestion, we have revised the title as follows:

Title: Impacts of Household Economic and Mothers’ Time Affluence on Obesity in Japanese Preschool Children: A Cross-sectional Study

2) Title: Suggest- in Japanese preschool children.

Response: As per the suggestion, we have revised the title as follows:

Title: Impacts of Household Economic and Mothers’ Time Affluence on Obesity in Japanese Preschool Children: A Cross-sectional Study

3) Line 12-13: It is unclear how household economic and mothers’ time affluence were used? Did the authors combine them in one variable? Household economic should be clearly defied in abstract.

Response: For household economic affluence and mothers' time affluence; we have included the following explanation in the revised manuscript to clarify how the variables were used.

Line14: Subjective household economic or mothers’ time affluence was divided into "affluent,” "neither,” "less affluent,” and "non-affluent" groups.

4) Line 15: What do you mean by “meta-analysis”? This is a research study not a review paper.

Response: The meta-analysis employed in this study has been used in other published articles as well. Please refer to the following papers:

1) Borgi L, Rimm E B, Willett W C, Forman J P. Potato intake and incidence of hypertension: results from three prospective US cohort studies BMJ 2016; 353: i2351 doi:10.1136/bmj.i2351

2) Hu Y, Ding M, Sampson L, Willett W C, Manson J E, Wang M et al. Intake of whole grain foods and risk of type 2 diabetes: results from three prospective cohort studies BMJ 2020; 370 :m2206 doi:10.1136/bmj.m2206

3) Zhou, J., Kelsey, K. T., Giovannucci, E., & Michaud, D. S. (2014). Fluid intake and risk of bladder cancer in the Nurses' Health Studies. International journal of cancer, 135(5), 1229–1237. https://doi.org/10.1002/ijc.28764

5) Line 21-23: Please include the OR and P-value only.

Response: Since the guidelines (see below link) have indicated that it is important to make decisions based on confidence intervals rather than p-values, we have included ORs and 95% CIs in the present study.

New Guidelines for Statistical Reporting in the Journal | NEJM(https://www.nejm.org/doi/full/10.1056/nejme1906559)

6) In abstract, please mention the type of design (e.g., cross-sectional, longitudinal) and the age of participants. Also, the conclusion is highly unacceptable and it does not propose a clear direction for future studies.

Response: We thank the reviewer for the comment. In the revised manuscript, we have included the study design in the title. We have also included the age of participants and future research directions.

Title: Impacts of Household Economic and Mothers’ Time Affluence on Obesity in Japanese Preschool Children: A Cross-sectional Study

Line13: The target population was 2-6-year-old preschool children and their mothers.

Line370: Mediators, including nutritional factors, need to be clarified in future studies. It might be possible to develop measures to improve the obesity problem in children by analyzing and managing mediators.

7) Please include “Japan” in the keywords list.

Response: We have included “Japan” in the keywords.

8) The introduction is too brief and should be strengthened with more references. There are only 13 references with 19 references used for the whole paper. The section needs to be expanded in my opinion. What is the labour force participation rate of women in Japan? The literature should also be in more depth and discuss the associations between work status and mother's and children’s dietary intake/obesity. For example, working part-time and/or full-time were associated with time-related barriers to healthful eating among adult men/women (Am J Health Behav. 2012 Nov; 36(6): 786-796) and their children (International Journal of Consumer Studies 2018, 42(5): 522-532).

Response: We thank the reviewer for the comment. In the introduction section of the revised manuscript, we have included sentences and references related to the working status of mothers, lack of time, and child obesity.

Line51: Furthermore, it has been suggested that the long working hours of mothers im-pact the overweight status of young children [16, 17]. Maternal employment is associated with less time spent caring for children and cooking for them; nota-bly, this difference in time (maternal working hours and caring for children) is the highest among mothers with young children (aged 0–5 years) [18]. Adult who work part-time and/or full-time have time-related barriers to healthy eating [19]. Since children eat food prepared by their parents, lack of household time may lead to childhood obesity.

9) Line 40-41: Please explain how mothers’ time affluence is a significant mediating factor between SES and health?

Response: We noted that the wording of this sentence was confusing; hence, we have removed it from the revised manuscript.

10) Line 41-46: Is there any evidence in Japan or other Asian countries?

Response: To our knowledge, there is no such report in Asia.

11) Line 47-51: The novelty of this paper is very weak in my opinion. The associations between household economic affluence and children's obesity should be clarified by providing more evidence from other countries. Why this study is important to conduct in Japan? Why not in other Asian countries? Why preschool children?

Response: From a life course perspective, there is great significance in focusing on children's health issues. However, research on infants and toddlers is still scarce. Given that Japan has one of the highest relative poverty rates among OECD countries, evaluating the association between household economic affluence and childhood obesity using data from Japanese data might provide a clearer view of these associations. In addition, Japan is a country with a large gender gap, and it has been pointed out that there is a large difference in the time men and women spend on housework and childcare. This can lead to a loss of time and leisure time, especially for working women. Therefore, the results of this Japanese study are applicable mainly in high-income countries.

In light of the above, we have added the following sentences:

Line38: Furthermore, childhood obesity is associated with obesity risk in adulthood [7, 8]. Previous studies have also demonstrated a strong consistent relationship be-tween low SES in early life and increased obesity in adulthood [9-11]. Childhood obesity is associated with increased early mortality and risk of later cardiometabolic morbidity (diabetes, hypertension, ischemic heart disease, and stroke) in adult life [12]. Although childhood obesity and SES determine health through-out life, studies on infants and toddlers are still scarce worldwide.

12) Line 69-74: Please describe in much more details how children were recruited?

Response: We have added the following description of the recruitment process:

Line76: Potential participants of the NNSPC were children aged <6 years (born from June 01, 2009, to May 31, 2015; approximately 5.5 thousand children) and their households (approximately 4.4 thousand households) in the 1,106 districts, ran-domly selected from the districts set by the National Survey of Living Standards, an institutional statistical survey conducted by the Ministry of Health, Labour and Welfare of Japan.

13) Line 78-82: This is not a good reason for exclusion men from analysis. This needs to be mentioned as a limitation of the study.

Response: As per the reviewer’s recommendation, we have included a statement in the limitations section that excluding men was a limitation.

Line362: Third, this study included only mothers. In the previous study, both mothers’ and fathers’ long working hours were associated with young children’s over-weight [16]. Therefore, in the future, it will be necessary to collect information from both mothers and fathers.

14) Figure 1 should be re-organized.

Response: We have re-organized Figure 1 for better visibility in the revised manuscript.

15) Line 106-109: This is not true. Why affluent and neither are combined in one group? This is not logic in my opinion.

Response: We set up this group based on previous studies as follows:

Tomata Y, Tanno K, Zhang S, Sakai M, Kobayashi K, Kurasawa N, Tanaka M, Kamada Y, Tsuji I, Hiramoto F. Subjective Household Economic Status and Obesity in Toddlers: A Cross-Sectional Study of Daycare Centers in Japan. J Epidemiol. 2019 Jan 5;29(1):33-37. doi: 10.2188/jea.JE20170081. Epub 2018 Jun 9. PMID: 29887543; PMCID: PMC6290275.

As previous studies have suggested the possibility as follows, it is common for modern Japanese people to respond that the "neither more nor less"  may be closer to the meaning of "yes".

Tatsuya Ohno, Wataru Noguchi, Yuko Nakayama, Shingo Kato, Hirohiko Tsujii, and Yoshihiko Suzuki. How Do We Interpret the Answer "Neither" When Physicians Ask Patients with Cancer "Are You Depressed or Not?". Journal of Palliative Medicine.Aug 2006.861-865.http://doi.org/10.1089/jpm.2006.9.861

16) Line 130: A large number of covariates were used, so there is a high change for bias during analysis.

Response: We thank the reviewer for the comment. We realized that in the initial submission, the section names were confusing and decided to change them: we have included them separately under the “other items” category. These variables were only introduced in the respondents’ characteristics in Tables 1 and 2, and not all of them were used in the multivariate adjustment.

Line136: 2.5. Other items

17) Line 202: than the "affluent" group?

Response: For clarity, in the revised manuscript, we have rephrased the sentence.

Line207: In the category of household composition, the "non-affluent" group had a higher percentage of single-mother households, a lower percentage of full-time em-ployees, a higher percentage of "do not ask for childcare”, and a higher per-centage of lack of exercise than the other ("affluent", “neither’, “less affluent”) groups.

18) I believe Chi-square and ANOVA test should be used in Table 1 & 2. Please include an additional column to show a significant P-value between the four groups.

Response: We used Chi-square and ANOVA in Table 1 & 2. As per the reviewer’s suggestion, we have included P-values between the four groups in Table 1 & 2 in the revised manuscript.

19) Please combine Table 3 & 4 in one table.

Response: In the revised manuscript, we have combined Tables 3 and 4 into one Table.

20) Line 265-276: This section doesn’t add any meaningful value to the paper. Please delete. Response: We think that pooling our results with previous studies would allow us to show that the results are informative. Further, it will also indicate a consistent association between household economic affluence and child obesity.

21) Line 278-287; 324-327: Please do not repeat results in the discussion.

Response: In the revised manuscript, we have removed results from the discussion section.

Line282: In the present cross-sectional study, we investigated the association be-tween household economic or mothers’ time affluence and obesity in preschool children. In an analysis of the data from the present study alone, lack of house-hold economic affluence was not associated with obesity in children; however, the OR in the “non-affluent” group tended to be higher compared to that in the "affluent" group.

22) The discussion is short and very weak in its current format. I suggest the authors to link the literature with the important results of this study. The implications of results and the directions for future research should be clearly discussed in conclusion.

Response: As per the reviewer’s suggestion, we have included a more detailed discussion and revised the conclusions of the revised manuscript.

Line339: A previous study reported that mothers working for long hours was a risk factor or overweight and obesity among preschool children, but this was mainly found in high-income families [16, 17]. Thus, the risk factors for obesity due to lack of household economic affluence and the risk factors for obesity due to lack of time affluence might be different.

Line356: Second, since all variables in the present study were subjective data, it is possible that a certain of random error (non-differential misclassification) leading to the underestimation of the result. Since previous studies have reported that mothers are more likely to overestimate their children's height and underesti-mate their weight [27]. In addition, the present study used only subjective data on time affluence, and was not able to examine actual time data. Future research using objective data is required. Third, this study included only mothers. In the previous study, both mothers’ and fathers’ long working hours were associated with young children’s overweight [16]. Therefore, in the future, it will be necessary to collect information from both mothers and fathers.

Line370: Mediators, including nutritional factors, need to be clarified in future studies. It might be possible to develop measures to improve the obesity problem in children by analyzing and managing mediators.

23) Line 327-331: Delete.

Response: We have retained this for the same reason indicated in response to comment 20.

24) Line 343-350: This section should be expanded. Please add additional limitations related to the study methodology and the statistical analysis used.

Response: The limitations section has been revised as per the reviewer’s recommendation.

Line350: First, the results of the present study might be underestimated due to selec-tion bias, although this study used a nationally representative sample. Previous-ly [26], it has been reported that individuals with lower annual household in-comes have lower response rates to the NNSPC. This association persisted after adjusting for potential confounders. Thus, the results of the present study might be underestimated because we focused on parents with “less-“ or “non-affluence”.

Line356: Second, since all variables in the present study were subjective data, it is possi-ble that a certain of random error (non-differential misclassification) leading to the underestimation of the result. Since previous studies have reported that mothers are more likely to overestimate their children's height and underesti-mate their weight [27]. In addition, the present study used only subjective data on time affluence, and was not able to examine actual time data. Future research using objective data is required. Third, this study included only mothers. In the previous study, both mothers’ and fathers’ long working hours were associated with young children’s overweight [16]. Therefore, in the future, it will be neces-sary to collect information from both mothers and fathers.

25) The manuscript requires significant editing by a native English speaker.

Response: In accordance with the reviewer's suggestion, we asked a native English speaker to proofread the paper

Reviewer 2 Report

First of all, I would like to congratulate the authors for choosing a novel and globally relevant topic of study such as childhood obesity.

A number of comments for the improvement of the manuscript are provided below.

Some parts of the methodology should be modified, as they are taken word for word from another already published article ''Subjective Household Economic Status and Obesity in Toddlers: A Cross-Sectional Study of Daycare Centers in Japan'' (study design).

It would be interesting if the authors had made statistics on male fathers and their attendance time in order to compare them with those of mothers.

In the discussion there are few citations, it would be good to increase them to give more quality to the manuscript.

Author Response

RESPONSE TO REVIEWER 2:

First of all, I would like to congratulate the authors for choosing a novel and globally relevant topic of study such as childhood obesity.

A number of comments for the improvement of the manuscript are provided below.

Response: We sincerely appreciate the time you have taken to carefully review our manuscript and provide valuable input.

1) Some parts of the methodology should be modified, as they are taken word for word from another already published article ''Subjective Household Economic Status and Obesity in Toddlers: A Cross-Sectional Study of Daycare Centers in Japan'' (study design).

Response: As per the reviewer’s recommendation, we have revised the methodology as follows. 

Line76: Potential participants of the NNSPC were children aged under 6 years (born from June 01, 2009, to May 31, 2015; approximately 5.5 thousand children) and their households (approximately 4.4 thousand households) in the 1,106 districts. They were randomly selected from the districts set by the National Survey of Living Standards, an institutional statistical survey conducted by the Ministry of Health, Labour and Welfare of Japan. Trained investigators visited each potential household once in September 2015 and distributed a self-administered questionnaire (for each child) to the mother or guardian who was usually in charge of meal preparation.

2) It would be interesting if the authors had made statistics on male fathers and their attendance time in order to compare them with those of mothers.

Response: Thank you for your suggestion. Because the NNSPC included a question about mothers' time, but not about fathers, we were unable to examine fathers' time enrichment. We have considered this as a limitation of the study.

Line362: Third, this study included only mothers. In the previous study, both mothers’ and fathers’ long working hours were associated with young children’s over-weight [16]. Therefore, in the future, it will be necessary to collect information from both mothers and fathers.

3) In the discussion there are few citations, it would be good to increase them to give more quality to the manuscript.

Response: As per the reviewer’s recommendation, we have included more citations in the discussion section.

Line339: A previous study reported that mothers working for long hours was a risk fac-tor for overweight and obesity among preschool children, but this was mainly found in high-income families [16, 17]. Thus, the risk factors for obesity due to lack of household economic affluence and the risk factors for obesity due to lack of time affluence might be different.

Line356: Second, since all variables in the present study were subjective data, it is possi-ble that a certain of random error (non-differential misclassification) leading to the underestimation of the result. Since previous studies have reported that mothers are more likely to overestimate their children's height and underestimate their weight [27]. In addition, the present study used only subjective data on time affluence, and was not able to examine actual time data. Future research using objective data is required. Third, this study included only mothers. In the previous study, both mothers’ and fathers’ long working hours were associated with young children’s overweight [16]. Therefore, in the future, it will be necessary to collect information from both mothers and fathers.

Reviewer 3 Report

REVIEW

Impacts of Household Economic or Mothers’ Time Affluence 2 on Obesity in Preschool Children 3 Kotone Tanaka 1, Kanami Tsuno 2 and Yasutake Tomata 1, *

Summary

This is a crossectional study that examines the impact of the availability of time by mothers on obesity I Preschool children in Japan.

From the analysis of the data from this study, the lack of household economic affluence was not significantly associated with obesity in their children, but the odds ratio in the group of “non-affluent” tended to be higher compared to " affluent" (OR, 1.67 [95% CI, 0.92-3.03]. Based on these results it is clear that a severe lack of household economic affluence was significantly associated with obesity in children in this group.

 Obesity frequency has a tendency to be higher among children whose mothers lack time affluent. However, the frequency of obesity was not higher when household economic affluent and mothers’ time affluent are combined together as two factors

Strength

Well a written manuscript, the English Grammar could be better. I will suggest that the author review this with someone whose first language is English

Weakness

The impact of men as a contributing factor even though the spend less time with children should not be ignored entirely.

The issues about affluence are best addressed with quantitative data rather than qualitative data.

Household income would have been a better parameter for the assessment of affluence.

I will suggest also using quantitative data set for the time affluence. Do the mothers in the study have different time affluence? If they do, what are the numerical values for this?

Author Response

RESPONSE TO REVIEWER 3:

REVIEW

Impacts of Household Economic or Mothers’ Time Affluence 2 on Obesity in Preschool Children 3 Kotone Tanaka 1, Kanami Tsuno 2 and Yasutake Tomata 1, *

Summary

This is a crossectional study that examines the impact of the availability of time by mothers on obesity I Preschool children in Japan.

From the analysis of the data from this study, the lack of household economic affluence was not significantly associated with obesity in their children, but the odds ratio in the group of “non-affluent” tended to be higher compared to " affluent" (OR, 1.67 [95% CI, 0.92-3.03]. Based on these results it is clear that a severe lack of household economic affluence was significantly associated with obesity in children in this group.

Obesity frequency has a tendency to be higher among children whose mothers lack time affluent. However, the frequency of obesity was not higher when household economic affluent and mothers’ time affluent are combined together as two factors

Response: We sincerely appreciate the time you have taken to carefully review our manuscript and provide valuable input.

1) Strength

Well a written manuscript, the English Grammar could be better. I will suggest that the author review this with someone whose first language is English

Response: The revised manuscript has been edited and proofread by a native English speaker.

2) Weakness

The impact of men as a contributing factor even though the spend less time with children should not be ignored entirely.

The issues about affluence are best addressed with quantitative data rather than qualitative data.

Household income would have been a better parameter for the assessment of affluence.

I will suggest also using quantitative data set for the time affluence. Do the mothers in the study have different time affluence? If they do, what are the numerical values for this?

Response: Thank you for your suggestion. Because the NNSPC included a question about mothers' time, but not about fathers, we were unable to examine fathers' time enrichment. In addition, the present study used only subjective data on time affluence, and was not able to examine actual time data. In the revised manuscript, we have included these points as limitations of the study.

Line356: Second, since all variables in the present study were subjective data, it is possible that a certain of random error (non-differential misclassification) leading to the underestimation of the result. Since previous studies have reported that mothers are more likely to overestimate their children's height and underestimate their weight [27]. In addition, the present study used only subjective data on time affluence, and was not able to examine actual time data. Future research using objective data is required.

Line370: Third, this study included only mothers. In the previous study, both mothers’ and fathers’ long working hours were associated with young children’s over-weight [16]. Therefore, in the future, it will be necessary to collect information from both mothers and fathers.

Round 2

Reviewer 1 Report

Dear Authors, 

The paper still needs more work. Many of my comments are not addressed.

1. 27 references are only used in the paper. The introduction and discussion should be strengthen with more references. I believe many studies are conduced on such topic.

2. The introduction could benefit from including the following: What is the labour force participation rate of women/men in Japan? What percentage of children go to childcare in Japan? What is the most common type of childcare in Japan? What is the impact of maternal employment on children obesity-associated dietary intake? Several studies are discussed in such point. For example: "Maternal work and children diet, activity and obesity"; "Parental work status and children dietary consumption: Australian evidence"; "The impact of maternal employment on children weight: Evidence from the UK"; Relationship between maternal employment status and children food intake in Japan"; "Examining the relationship between maternal employment and health behaviours in 5-year-old British children".

3. Please include your response on comment 11" From a life course perspective.....etc" at the end of introduction.

4. Clear rationale why affluent and neither are combined in one group. This should clearly mentioned.

5. Please change "other items" section to "confounding variables", and justify why each variable was used as a covariate.

6. The statistical analysis section should described each test used, including Chi-square and ANOVA.

7. I still believe section 3.4 is not needed. Clear rationale why this section should be included. Otherwise, please delete.

8. The discussion is still very weak and short. Please link the literature with the results of this study.

9. The conclusion section should be expanded. The implications of results and the directions for future research should be clearly discussed in this section.

10. English language usage should be improved in this paper.

Author Response

RESPONSE TO REVIEWER 1:

We sincerely appreciate the time you have taken to carefully review our manuscript and provide valuable input.

  1. 27 references are only used in the paper. The introduction and discussion should be strengthen with more references. I believe many studies are conducted on such topic.

Response: In response to comments 2 and 3, we have expanded the introduction section. In response to comments 7 and 8, we have expanded the discussion section.

  1. The introduction could benefit from including the following: What is the labour force participation rate of women/men in Japan? What percentage of children go to childcare in Japan? What is the most common type of childcare in Japan? What is the impact of maternal employment on children obesity-associated dietary intake? Several studies are discussed in such point. For example: "Maternal work and children diet, activity and obesity"; "Parental work status and children dietary consumption: Australian evidence"; "The impact of maternal employment on children weight: Evidence from the UK"; Relationship between maternal employment status and children food intake in Japan"; "Examining the relationship between maternal employment and health behaviours in 5-year-old British children".

Response: We thank the reviewer for this comment. As per this recommendation, we have added these sentences. In addition, we have also cited the references suggested by the reviewer.

Line47: It has been suggested that households with mothers who work as full-time employees or for long hours are associated with children having unhealthy eating habits and food intake [15-18]. These households are also associated with childhood obesity [15, 17-20].

Line53: Therefore, it is expected that the mother's lack of time affluence due to employment and other factors is associated with children's obesity through factors such as diet.

Line74: In Japan, "dual-income households” are trending upward. The Gender Equality Bureau, Cabinet Office, Government of Japan reported that the number of dual-income households that consisted of an employed husband and a wife without paid work (age of wife ≤64 years) was 718 households in 1985; however, it increased to 1177 households by 2021 (latest data) [25]. Moreover, Japan is a country with a large gender gap, ranking 116th out of 146 countries [26]. According to the OECD, women cared for household members and did routine housework 2.3 times longer than men on average; in Japan, this is 3.7 times longer [27]. Thus, Japanese mothers tend to have less time affluence for other time of working.

Line328: The utilization rate of daycare centers for 4-year-olds in Japan was 42.2% in 2018, according to the latest available data (reported by Ministry of Education, Culture, Sports, Science and Technology in Japan, 2021).

  1. Please include your response on comment 11" From a life course perspective.....etc" at the end of introduction.

Response: As per the reviewer’s recommendation, we have added these sentences.

Line39: From a life course perspective, focusing on children's health issues could be of great significance, but research on infants and toddlers is still scarce worldwide.

Line74: In Japan, "dual-income households” are trending upward. The Gender Equality Bureau, Cabinet Office, Government of Japan reported that the number of dual-income households that consisted of an employed husband and a wife without paid work (age of wife ≤64 years) were 718 households in 1985; however, it increased to 1177 households by 2021 (latest data) [25]. Moreover, Japan is a country with a large gender gap, ranking 116th out of 146 countries [26]. Ac-cording to the OECD, women cared for household members and did routine housework 2.3 times longer than men on average; in Japan, this is 3.7 times longer [27]. Thus, Japanese mothers tend to have less time affluence for the other time of working.

  1. Clear rationale why affluent and neither are combined in one group. This should clearly mentioned.

Response: In our previous study, the risk of childhood obesity was higher only in the "non-affluent" group, suggesting that household economic affluence and childhood obesity may not be a dose-response relationship. In the present study, we took this into account and included “neither” in the affluent group.

Line198: In our previous study [22], the risk of childhood obesity was higher only in the "non-affluent" group, suggesting that household economic affluence and childhood obesity may not be a dose-response relationship.

  1. Please change "other items" section to "confounding variables", and justify why each variable was used as a covariate.

Response: Since "other items" includes items that are not considered to be confounding variables, we have divided the section into "other items" and "confounding variables”. Additionally, we have added references to support the decision to define them as confounding variables.

Line157: 2.5. Confounding variables

The following items used responses obtained by a mother-reported questionnaire: age (months), sex, birth weight [30-32], and mother's age (years) [33-35].

Line161:2.6. Other items

The total number of siblings of children was obtained through a mother-reported questionnaire.

  1. The statistical analysis section should described each test used, including Chi-square and ANOVA.

Response: As per the reviewer’s recommendation, we have added these sentences.

Line188: Intergroup comparisons were performed using one‐way analysis of variance (ANOVA) for continuous variables and the χ2 test for categorical variables.

  1. I still believe section 3.4 is not needed. Clear rationale why this section should be included. Otherwise, please delete.

Response: We have added the rationale for the use of a meta-analysis in the present study as follows:

Line16: A logistic regression model and a meta-analysis of an earlier study of 1848 children in Japan were conducted to examine the association between household economic affluence and children’s obesity to estimate more plausible point estimates.

Line56: We have previously shown that household economic affluence and mothers’ time affluence are associated with obesity among preschool children aged 4 years old. Additionally, when mothers' lack of time affluence is added to household economic affluence, the impact of the latter on children's obesity was reinforced [22]. This previous study [22] was conducted using the same questions of household economic affluence from the National Nutrition Survey on Preschool Children (NNSPC). However, it [22] was conducted in only one region in Japan and only among children attending daycare centers; thus, it may not be a highly representative sample of Japanese pre-school children, and research with national representative data is needed.

Line84: The present study aimed to determine the association between household eco-nomic or mothers’ time affluence and obesity in children with national representative data from the NNSPC. We also aimed to conduct a meta-analysis to integrate the result of the previous study [22] in order to estimate more plausible point estimates.

Line210:                 A previous study we conducted [22] on 1848 4-year-old preschool children used the same questionnaire from the NNSPC employed in the present study [22]. In our previous study, household economic affluence was classified using the same definition as in the present study. The definition of obesity was also the same as in the present study, which was defined based on the International Obesity Task Force. To evaluate the heterogeneity of the results, a meta-analysis was conducted to examine the association between household economic affluence and children's obesity based on the combined results of our previous study.

Line320:                 In our previous study [22], the OR of children's obesity in the household economic "non-affluent" group compared with the "affluent" group was 2.31; however, in the present study, it was 1.68. There are two possible explanations for these discrepancies. First, there is a possibility of a difference in the study population. The study population of the previous study included only children attending a day-care center, whereas the present study included a nationally representative sample (not only including children attending a day-care center, but also children attending a kindergarten, being cared for by grandparents or relatives, and others). The utilization rate of daycare centers for 4-year-olds in Japan was 42.2% in 2018, according to the latest available data (reported by Ministry of Education, Culture, Sports, Science and Technology in Japan, 2021). The Act of Child Welfare Law in Japan ensures that daycare centers in Japan provide care for infants and toddlers who need it for various reasons, such as having working parents. Therefore, the participants in the previous study were more likely to be less affluent in their lives than those in the present study. However, the percentage of obese children in each category of household eco-nomic affluence ("affluent", "neither", "less affluent," and "non-affluent”) in the previous study were 6.2%, 7.3%, 5.4%, and 11.5%, respectively, whereas in the present study, they were 5.6%, 6.2 %, 5.7%, and 9.0%, respectively. Since there was not much difference in each group, it is unlikely that there was an underestimation due to selection bias. Second, there is a possibility of larger random errors due to the use of mothers-reported children’s weight and height in this study. Previous studies have reported that mothers are more likely to overestimate their children's height and underestimate their weight [36]. Thus, underestimation of association due to larger random error (non-differential misclassification) might have occurred because the previous study used measurements of weight and height taken by daycare centers’ staff. The OR was higher, especially in the “non-affluent” group in the adjusted model in the previous study (OR, 2.31 [95% CI, 1.234.33]). Conversely, the OR was higher in the “non-affluent” group in the present study (1.68 [0.93-3.03]). Thus, a meta-analysis focused on the prevalence of children’s obesity in the “non-affluent” group compared to the “affluent” was conducted to evaluate the heterogeneity. As a result, we found a significant association between household economic affluence and children’s obesity (OR, 1.95 [95% CI, 1.27-3.00]). The heterogeneity of the results of the meta-analysis was 0%. This suggests that the results of the present study would be considered consistent with those of the previous study [22].

  1. The discussion is still very weak and short. Please link the literature with the results of this study.

Response: As per the reviewer’s recommendation, we have expanded the discussion section as follows:

Line320:                 In our previous study [22], the OR of children's obesity in the household economic "non-affluent" group compared with the "affluent" group was 2.31; however, in the present study, it was 1.68. There are two possible explanations for these discrepancies. First, there is a possibility of a difference in the study population. The study population of the previous study included only children attending a day-care center, whereas the present study included a nationally representative sample (not only including children attending a day-care center, but also children attending a kindergarten, being cared for by grandparents or relatives, and others). The utilization rate of daycare centers for 4-year-olds in Japan was 42.2% in 2018, according to the latest available data (reported by Ministry of Education, Culture, Sports, Science and Technology in Japan, 2021). The Act of Child Welfare Law in Japan ensures that daycare centers in Japan pro-vide care for infants and toddlers who need it for various reasons, such as having working parents. Therefore, the participants in the previous study were more likely to be less affluent in their lives than those in the present study. However, the percentage of obese children in each category of household eco-nomic affluence ("affluent", "neither", "less affluent," and "non-affluent”) in the previous study were 6.2%, 7.3%, 5.4%, and 11.5%, respectively, whereas in the present study, they were 5.6%, 6.2 %, 5.7%, and 9.0%, respectively. Since there was not much difference in each group, it is unlikely that there was an underestimation due to selection bias. Second, there is a possibility of larger random errors due to the use of mothers-reported children’s weight and height in this study. Previous studies have reported that mothers are more likely to overestimate their children's height and underestimate their weight [36]. Thus, underestimation of association due to larger random error (non-differential misclassification) might have occurred because the previous study used measurements of weight and height taken by daycare centers’ staff. The OR was higher, especially in the “non-affluent” group in the adjusted model in the previous study (OR, 2.31 [95% CI, 1.234.33]). Conversely, the OR was higher in the “non-affluent” group in the present study (1.68 [0.93-3.03]). Thus, a meta-analysis focused on the prevalence of children’s obesity in the “non-affluent” group compared to the “affluent” was conducted to evaluate the heterogeneity. As a result, we found a significant association between household economic affluence and children’s obesity (OR, 1.95 [95% CI, 1.27-3.00]). The heterogeneity of the results of the meta-analysis was 0%. This suggests that the results of the present study would be considered consistent with those of the previous study [22].

Line356: We observed that lower mothers’ time affluence tended to be inversely as-sociated with the prevalence of obesity in the present study. Although we have not found any previous studies directly examining the association between mothers' lack of time affluence and children's obesity, the results of the present study on mothers' time affluence may be logically consistent with previous findings. For example, the previous cohort study showed that households with mothers who were employed as full-time workers were associated with a higher BMI and thus had children with excess weight [19]. The results of the present study showed that the percentage of full-time employees was higher in participants who responded "not affluent" of mothers’ time affluence. The interpretation that mothers' lack of time affluence is associated with childhood obesity would not be logically inconsistent if full-time employment makes mothers lose time affluence more easily.

Line375: It is possible that mothers' lack of time affluence may affect their children's obesity via unhealthy dietary habits.

  1. The conclusion section should be expanded. The implications of results and the directions for future research should be clearly discussed in this section.

Response: As per the reviewer’s recommendation, we have added the implications of the results and directions for future research in the conclusion section as follows:

Line408: The frequency of obesity tended to be higher among children in households lacking economic affluence and with mothers who lacked time affluence. How-ever, this frequency was not synergistically higher when household economic affluence and mothers’ time affluence were combined. Future research using objective data of household economic affluence and time affluence is required. In addition, mediators, including nutritional factors, need to be clarified in future studies. It might be possible to develop measures to improve the issue of childhood obesity by analyzing and managing mediators.

  1. English language usage should be improved in this paper.

Response: In accordance with the reviewer's suggestion, we asked a native English speaker to proofread the paper again.
